# Comparing Protein Language Models Using Remote Homology Detection for Phages

## Abstract

**Background.** Protein language models (pLMs) are machine learning models that learn high-dimensional representations of protein sequences. These models have utility in biological settings; pLMs can convert between protein sequence and structure (Heinzinger et al., 2023), determine evolutionary relationships between organisms (Bordin et al., 2023), and design protein sequences with desired functions (Madani et al., 2023). Transfer learning with previously trained pLMs offers a powerful, minimal resource strategy for performing diverse large-scale classification and prediction tasks. However, as pLMs proliferate in the research community with differences in training objectives, model structure(s) and training datasets, it is daunting for a less-experienced end user to decide which pLM to use for biological experiments and discovery. Consequently, it is essential to compare pLMs to determine their strengths and limitations in use-cases relevant to biological researchers. Here, we present a comparison of the performance of pre-trained pLMs in a difficult remote homology detection task for phage proteins described previously in Flamholz et al. (2024). We make available our code and notebooks to facilitate other research scientists to use such models via anonymous Github `https://anonymous.4open.science/r/plm-model-comparison-7733/README.md`. **Results.** Variations in model training resulted in significantly different performance in our biological task. We present an analysis that compares five recently published pLMs : (1) ProtT5, (2) ProstT5, (3) TMVec, (4) ESM-2, and (5) CARP. We observed that all models were able to capture information that could be used to annotate viral proteins. Model embeddings could be used to train functional classifiers that, when tested using the large PHROG and EFAM databases of phage proteins, captured meaningful biological information. Performances across models were noticeably different for this task. Models trained on larger, more diverse databases of genomic sequences, such as Big Fantastic Database (BFD), performed better overall. Models with Transformer architectures performed better than those with the convolutional neural network (CNN) architectures. **Conclusion.** The utility of pLMs in areas of biological research is clear; we demonstrate such models are useful for remote homology detection in phage genomes, an area of active interest in environmental and clinical biology. Our study provides a framework for how biological scientists can choose pLMs to incorporate into their experiments and analyses. Overall, while some models clearly performed better, on the whole, all pLMs achieved high scores for prediction. For end-users, the implication is that many pLM models are useful, but domain knowledge coupled with specialized model training paradigms may improve results when addressing specific biological questions.

## 1 Background

Natural language processing (NLP) algorithms are algorithms that model language by converting text into numerical representations that capture information about the context and meaning of words. Researchers have used NLP algorithms on protein sequences to learn representations of amino acid (AA) sequences that capture biologically meaningful properties (Iuchi et al., 2021). The representations embed information related to protein structure, function, and classification. Representations

have also been shown to carry information about the relationship between different sequences. Protein language models (pLMs) are a subset of biologic NLP models that biologists can use to categorize protein sequences (Flamholz et al.). Protein sequence annotation is an unsolved and key problem for biological discovery and application, and pLMs may enable detection of relationships between proteins that are outside the capacity of current state-of-the-art approaches. pLMs are trained on large datasets of AA sequences and can capture the context provided by the position of AAs (CARP) (Yang et al., 2024), predict interactions between protein residues (Foldseek) (van Kempen et al., 2024), generate protein sequences (ProGen) (Madani et al., 2023), or taxonomize protein sequences (Genomic Language Model) (Hwang et al., 2024). Biologists have begun using these generalized models to formulate experimental hypotheses (Hie et al., 2023); however, many biologists still train models on small datasets for specific tasks and these models have not been widely adopted in the biological sciences.

Within the past decade, a host of different pLMs were developed. These models were trained on different sequence datasets with different model architectures, and were designed to perform a multitude of different tasks. Deciding which model is best to use in an experiment, especially for domain-specific tasks useful to individual scientists, can be daunting for a non-expert (Flamholz et al.). The comparison experiment described here, applying pLMs to annotating viral proteins in large, diverse, metagenomic datasets, will make these models more accessible to biologists.

Viruses are abundant, fundamental players in shaping life on Earth. Present in every environment from gut microbiomes to soil samples, viruses radically alter the genomes and populations of their host organisms. Bacteriophages, or phages that target bacteria specifically, have significant impacts on the microbial communities. Phages shift the dynamics between bacterial organisms, driving how the ecosystem functions (Kauffman et al., 2022). In ecosystems such as the gut microbiome, phage-bacterial interactions can trigger disease in the host, or can protect the organism against pathogenic bacteria (Zhang et al., 2023). This impact is in part due to the ability of phages to evolve rapidly alongside their bacterial hosts (Koskella et al., 2022).

Consequently, it is crucial to develop a robust taxonomy of phages in order to best understand and predict the impact of these interactions. However, due to the lack of conserved marker genes in viruses, thousands of viruses discovered in viral catalog studies go unclassified (Flamholz et al., 2024). The pace of viral genome discovery is also rising with environmental metagenomic sequencing (Kuhn, 2021), (Camargo et al., 2023), reinforcing the importance of innovative, accessible, solutions to this problem.

Current phage annotation methods include profile-profile Hidden Markov Model (HMM) and other sequence-based homology methods. However, these methods suffer from the limited amount of annotated viral protein sequences, costliness of sequence-based annotation and rapid rate of phage evolution. It is difficult to construct statistical models from poorly annotated datasets. Due to rapid evolution, annotating phages based on immediate evolutionary relationships is unfeasible. We showed previously that annotation of uncultivated phage genomes is aided by pre-trained pLMs (Flamholz et al., 2024) but the proliferation of pLMs in the community prompts the question of whether different training regimes influence the results, and if so, how.

In this comparison experiment, we present five pLMs that each have unique elements related to their training, structure and dataset. We show that each can produce protein representations that are useful for classification-based transfer learning, but that differences in training corpus and model architecture affect performance on our remote homology detection task. We test the performance of these models on two, large viral sequence databases, PHROGs (Terzian et al., 2021) and EFAM (Zayed et al., 2021).

## 2 DATA DESCRIPTION

Experiments were conducted using two phage sequence databases, Prokaryotic virus Remote Homologous Groups (PHROGs) and EFAM (Terzian et al., 2021), (Zayed et al., 2021). The PHROGs v4 database stores 868,340 protein sequences, clustered into 38,880 viral protein families (VPFs) using a novel method for remote homology detection. The protein sequences were first gathered using similarity searches, and then clustered into protein families using HMM profiles. 5,134 of the protein families were then annotated as belonging to one of nine functional categories us-

ing annotation transfer(Terzian et al., 2021). The EFAM database stores 240,311 Hidden Markov Model (HMM) profiles of VPFs, identified from the Global Ocean Virome 2.0 database. Each aligned cluster of viral proteins was assigned an annotation and a probability (Zayed et al., 2021). The PHROGs and EFAM databases were selected together over other viral databases because of their lack of overlap. The PHROGs database consists of known viral proteins and complete viral genomes, taken from viruses that infect Archaea and Bacteria. The EFAM database consists of higher-confidence viral contigs from the ocean and curated after the end date for sequence inclusion in PHROGs. The PHROGs database V4 `https://phrogs.lmge.uca.fr/` was downloaded on 12/03/2023. The EFAM database was downloaded from the project repository of Flamholz et al. (2024) on Github on 6/11/2024.

Five trained pLMs were used for this experiment. The ProtT5_XL_Uniref50 (Elnaggar et al., 2022), ProstT5 (Heinzinger et al., 2023) and Esm2_t30_150M_UR50D (Lin et al., 2023) models were accessed via Hugging Face. The CARP_640M model (Yang et al., 2024) was accessed via the sequence-models python package `https://github.com/microsoft/protein-sequence-models`. The TM-Vec model (Hamamsy et al., 2022) was accessed via the tm-vec python package `https://github.com/tymor22/tm-vec` (Table 1).

For each of the five models, the training dataset, number of parameters, number of layers in the model, embedding dimensions of the models, structure of the models and pre-training objectives were listed. These training strategies were of interest because they were hypothesized to have an effect on model performance in our experiment. Each of these pLMs were used to embed the entire PHROGs and EFAM databases.

## 3 MODELS

For each of the five models, training methods such as the dataset and structure were documented.

**Table 1.** Training strategies for each pLM

| Training Method | ProtT5-XL-Uniref50 | Esm2-t30-150M-UR50D | CARP-640M | TM-Vec CATH | ProstT5 |
|---|---|---|---|---|---|
| Dataset | Uniref50, BFD100 | Uniref50, Uniref90 | Uniref50 | CATH | AlphaFold Protein Structure Database |
| Number of Parameters | 3B | 150M | 640M | 17.3M | 17M |
| Number of Layers | 24 | 30 | 56 | – | – |
| Embedding Dimensions | 1024 | 640 | 1280 | 512 | 1024 |
| Structure | Encoder-Decoder transformer | BERT-style encoder only transformer | ByteNet dilated CNN | Transformer encoder, average pooling, dropout, fully connected layers | Encoder-Decoder Transformer |
| Training Objective | Span-based denoising | MLM | MLM | Minimize L1 distance between cosine similarities of pairs | Span-based denoising |

### 3.1 PROTT5

ProtT5-XL-Uniref50 is an example of one of the first pLMs. It was created as an example of how machine learning models can capture meaningful biological information from protein sequences alone, rather than evolutionary information, which is computationally costly and not always available. The model was trained on BFD100 (Jumper et al., 2021), and fine-tuned on Uniref50 (Suzek et al., 2015). It has an Encoder-Decoder Transformer structure. ProtT5 utilizes the same training ob-

jective as BERT, where single tokens were corrupted and reconstructed with masking probabilities of 15%.

## 3.2 ESM-2

ESM-2 was trained to learn large amounts of information and representations from protein sequences. The same model was trained on multiple scales, ranging from 8 million parameters to 15 billion parameters, making ESM-2 the largest model at the time of its release (Lin et al., 2023). The model was trained on the Uniref50 and Uniref90 databases, and has a Transformer architecture with an attention mechanism to learn pairwise interactions between amino acid sequences. ESM-2 has a masked language modeling (MLM) training objective, where 15% of amino acid tokens were hidden, and the model was tasked with predicting them.

## 3.3 CARP_640M

CARP is an example of a convolutional neural network (CNN)-based model, and was provided as an efficient alternative to the prevalent Transformer-based models in the market. The model was trained on sequences from Uniref50. CARP models were trained using the masked language modeling objective, where 15% of tokens from each sequence were randomly selected. 80% of these tokens were replaced with a mask token, 10% were replaced with a random amino acid, and 10% were unchanged.

## 3.4 TM-VEC CATH

TM-Vec was designed to predict the TM-score, a measure of structural similarity, between two protein sequences without the intermediate computation of their structures. The model was trained on sequences from the CATH and SwissModel structural databases. The training objective of TM-Vec was to reduce the L1 distance between the cosine similarity of the proteins' function-reduced representations and their TM-scores.

## 3.5 PROSTT5

ProstT5 was designed to translate between protein sequences and 3Di (structural) tokens. To create ProstT5, ProtT5 was fine-tuned on the AlphaFold protein structure database. The model shares the same structure and training objectives as ProtT5 (Encoder-Decoder Transformer and span-based denoising).

## 4 ANALYSES

### 4.1 PHROGS CLASSIFIER PERFORMANCES

PHROGs multi-class classifiers were trained on the embeddings from each model for five folds, following the procedure for training PHROGs classifiers from Flamholz et al. (2024). The novel classifiers were compared with the Transformer_BFD classifier trained in Flamholz et al. (2024). The average true positive rates and false positive rates over the five folds were graphed, and the average AUC and SD were calculated. Across all categories, the mean AUROCs were calculated (Figure 1a).

All five novel classifiers performed well (minimum AUROC was 0.91), with ProstT5 and ProtT5 performing the best with AUROCs of 0.92. The average precision and recall over the five folds were graphed, and the average AUC and SD were calculated (Figure 1b). Across all categories, the mean AUPRCs were calculated. ProtT5 performed the best, with an AUPRC of 0.72 and the original functional classifier performed the worst, with an AUPRC of 0.63, illustrating that these newer pLMs are superior to state-of-the-art tools for phage annotation.

The precision, recall and F1 scores of each of the classifiers were compared via boxplot (Figure 2). Across the categories, the models performed the best in the 'tail' and 'DNA, RNA and nucleotide metabolism.' CARP was the least performative, along with the original classifier trained in Flamholz et al. (2024). Across the categories, the models performed the best in 'lysis', 'tail', and 'DNA, RNA,

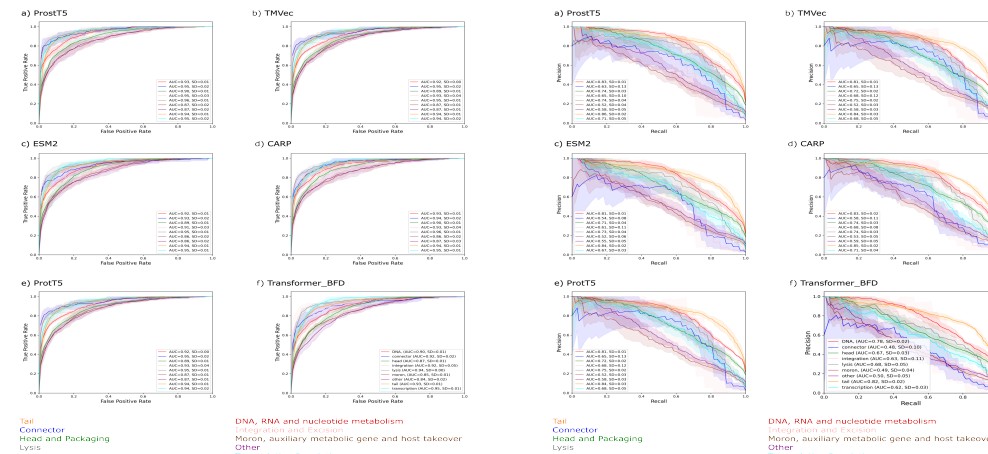

Figure 1a.                    Figure 1b.

**Figure 1.** Functional category classification using the PHROGs classifiers trained on pLM embeddings. For each fold, training was done on entire families. Testing was done on randomly selected sequences. Protein sequences were embedded using each of the five models. **Figure 1a.** PHROG Classifier ROC (Receiver Operator Characteristic Curve) performance over five folds, with per-category AUC and standard deviation (SD). The average AUROC across all categories for each model is stored in Supplemental Data Table 1. **Figure 1b.** PHROG Classifier PRC (Precision Recall Curve) performance, with per-category AUC and SD. The average AUPRC across all categories for each model is stored in Supplemental Data 1.

and nucleotide metabolism' categories. ProstT5, ProtT5 and TM-Vec performed the best by all three metrics. These three models had structural training objectives, indicating that the training objective has a significant influence on model performance.

## 4.2 EFAM CLASSIFIER PERFORMANCES

The CARP model embeddings were excluded due to its poorer performance, as illustrated by the trained PHROGs classifier performance (Figure 2). The EFAM multi-class classifiers were trained on the embeddings from each model for five folds, using the same training parameters as the PHROGs multi-class classifiers. "True" functional category predictions were assigned to the EFAM database itself using the predictions from Flamholz et al. (2024).

The precision-recall (Figure 3a) and F1-FDR (Figure 3b) curves indicated a strong performance across all categories for each of the models. All of the models had an average AUPRC across categories well above 0.9. However, the model calibration curves (Figure 4a) displayed overconfidence in all of the models, indicating possible overfitting.

We tested the functionality of the EFAM classifiers on a novel prediction task by using the classifiers to label EFAM families that were not annotated by PHROGs HMMs (Figure 5). The ProtT5, ProstT5, TM-Vec and ESM-2 classifiers expanded the annotated fraction of EFAM by 33.2%, 26.4%, 27.8% and 24.9%, respectively, with the most novel predictions made in the 'head and packaging,' 'tail' and 'DNA, RNA and nucleotide metabolism' functional categories. These results indicate that the generalized pLMs can supplement state-of-the-art HMMs in remote phage homology detection.

To demonstrate that the pLM-based classifiers can be applied to a specific question of biological interest, we examined the same 'integration and excisionase' category that Flamholz et al. (2024) examined (Supplemental Figure 3). This category was chosen due to its biological application to identifying temperate bacteriophages (Flamholz et al., 2024).

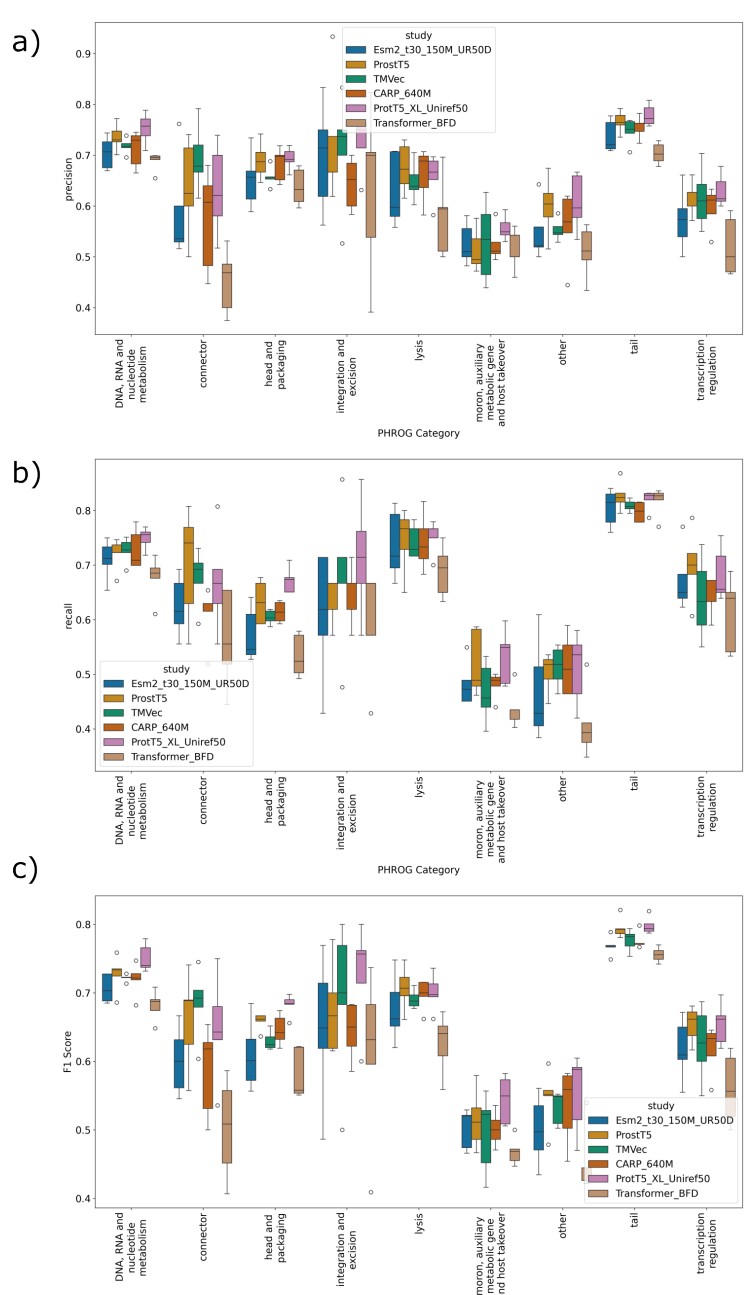

**Figure 2.** Boxplot comparisons of the five PHROGs functional classifiers. Performance is measured over five folds. The precision (Figure 2a), recall (Figure 2b) and F1 (Figure 2c) scores for each model were compared by category. Overall, the models performed the best in the tail, lysis, and DNA, RNA and nucleotide metabolism categories. Boxes represent interquartile range; horizontal line indicates median; whiskers indicate the entire distribution, with the exception of outliers (shown as circles).

## 5 DISCUSSION

Biologists have trained smaller task-specific models for their experiments, such as detecting abnormalities in the gastrointestinal tract (Rustam et al., 2021). However, training models on these datasets can lead to issues such as overfitting, where the model recognizes patterns in the dataset that are not generalizable, and overestimation of model performance. Here, we take a large biolog-

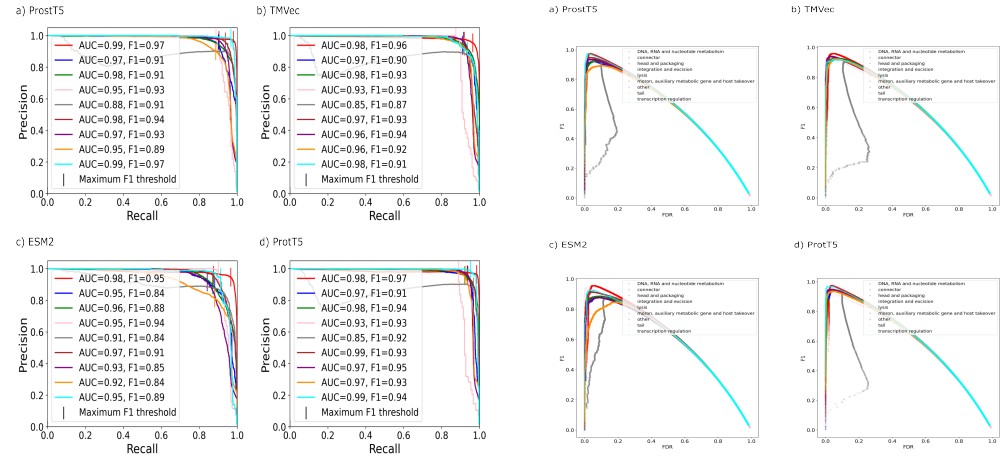

**Figure 3a.**                    **Figure 3b.**

**Figure 3.** EFAM Classifier Performance Validation. PHROG annotated EFAM families were used as ground truth for the predictions. **Figure 3a.** EFAM Classifier PRC Performance. Functional categories are scored using F1 and AUC. **Figure 3b.** EFAM Classifier F1 versus FDR Performance.

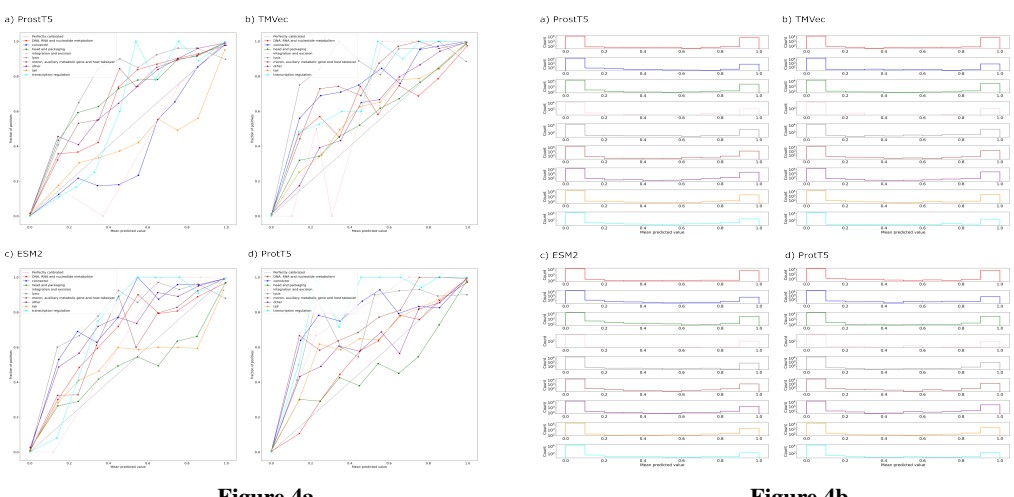

**Figure 4a.**                    **Figure 4b.**

EFAM Classifier Calibration Analysis. EFAM VPFs labeled by PHROG HMMs were used to test the model calibration over each category. **Figure 4a.** EFAM Classifier Calibration Curves. A perfectly calibrated model (where the mean predicted value is equivalent to the fraction of positive predictions) is represented by the gray dashed line. Graphs above the perfect model indicate overconfidence, while graphs below the perfect model indicate underconfidence. **Figure 4b.** Histograms showing the distribution of predictions across the test set for each category, for each probability.

ical problem of general interest: annotation of distantly related viral proteins, and demonstrate that classifiers trained on recent pLM embeddings perform significantly better than classifiers trained on older pLM embeddings.

Despite there being significant differences in the architecture and training of the pLMs tested here, the models performed relatively similarly. Functional classifiers trained on the models achieved high F1 and AUPRC scores when predicting the functional categories of PHROGs and EFAM families. The EFAM classifiers made novel predictions on the EFAM families, with annotation gains of 25 to 33 percent in this large database. Our results show that pLMs are powerful tools for reaching uninterrogated areas of annotation space in the unsolved problem of remote phage homology. Our

work suggests that models with large, diverse training datasets and structure-based objectives will perform the best for these tasks and should be prioritized for biological applications.

It is an ongoing debate whether the size of the training dataset for the model is the main contributor to model performance. Developers of large models from ESM-1, with 43 million parameters in 2019, to models scaling to the billions argue that the larger the training dataset, the better the model (Serrano et al., 2023). This comparison experiment indicates that these hypotheses are viable. ESM-2 was scaled down to 150M parameters, as the larger models caused memory out of limit errors; we note that this limitation is important for end-users who do not have access to computing resources that can utilize the full ESM-2 model. ESM-2 also performed the poorest out of all of the five models in multiple experiments.

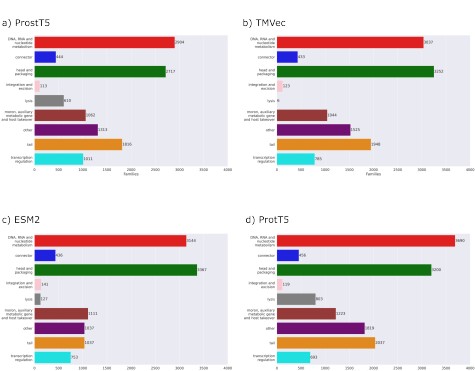

**Figure 5.** EFAM Classifier Predictions for VPF families not annotated by PHROGs HMM Profiles. The EFAM classifiers were used to make novel predictions for EFAM families that could not be labeled using the PHROGs HMM profiles. Families were annotated to the category-specific thresholds.

Our comparison methods additionally demonstrate that the content of the database and model architecture has an impact on the performance of the model on prediction tasks as well. BFD contains more sequence diversity than Uniref50 or Uniref90 as a genomic sequence database. Models trained or pre-trained on BFD, including ProtT5 and ProstT5 tended to perform better. Moreover, ProtT5 and ProstT5 have Transformer architectures, which is an extremely effective structure for pLMs, albeit computationally costly. CNN models such as CARP performed poorly in comparison.

Training objectives may also have an impact on model performance when these pLMs are applied to biologically informative tasks. PHROGs classifiers trained on embeddings from models with structural objectives such as ProstT5, ProtT5 and TM-Vec performed the best when models were compared. Models with structure-based objectives may perform better on annotation tasks compared to models with sequence-based objectives.

Scientists introducing novel models such as ESM-2 already compare different scales of their model (Lin et al., 2023), illustrating that larger model size has a positive impact on model performance. However, experiments comparing the impact of model structure or training dataset alone have not been conducted. Instead of using model scale as the independent variable when comparing multiple models, we can use model structure (training multiple models on the same set of parameters, and comparing their performances) or training dataset (training multiple models with the same structure on similarly sized sets of different parameters).

We hypothesize that for the remote homology prediction task here, performance is influenced by structure-based objectives due to the nature of viruses. Viruses evolve rapidly, however, certain structures such as the capsid protein are evolutionarily conserved. Will structure-based objectives be as useful when applying pLMs to remote homology prediction tasks in Bacterial, Archaeal, or Eukaryotic proteins? Will larger and more diverse training sets enable us to cover the large swaths of protein sequence space that we still cannot annotate? Answering such questions may begin to uncover fundamental rules of protein function across the domains of life.

When answering other biological questions such as predicting a person's susceptibility to disease or designing protein sequences for specific tasks, different training parameters may have greater impacts on model performance. We encourage end-users to use these powerful pLMs on other specific biological questions, testing their applicability and expanding domain knowledge.

## 6 METHODS

### 6.1 EMBEDDINGS

The sequence information from the PHROGs fasta files were uploaded into a Virtual Machine (VM) hosted by Google Cloud. The models were directly hosted on the VMs via Python 3 scripts. For each of the embedding experiments, a VM with a single NVIDIA L4 GPU under the G2 series, 16 vCPU, 8 cores and 64 GB of memory (g2-standard-16) was utilized. The operating system was Deep Learning with Linux, and the version was Deep Learning VM with CUDA 11.8 M123. The boot disks are balanced persistent, with sizes of 100 GB each. The average runtime for each experiment was two to three days, and batch sizes of 1 were used.

ProstT5, ProtT5 and ESM-2 were accessed via the Hugging Face Hub. CARP and TM-Vec were accessed via the sequence-models and tm-vec packages respectively. Models were run using the provided Python functions from their respective Github repositories. To create the averaged embeddings, the n-dimensional embeddings from each model were grouped based on their PHROG families. The arithmetic mean was taken across each column to create a single n-dimensional vector for each family.

### 6.2 TRAINED MODEL PERFORMANCES ON PHROGS

A multi-class classifier was trained on the embedded PHROGs database for each of the five model embeddings using the methods described in (Flamholz et al., 2024). The classifier architecture is a dense, feed-forward neural network. The models were trained using Tensorflow with the following parameters: loss=categorical_crossentropy, opt=Adam(0.0001), batch_size=60. The networks of the models had three hidden layers each, with the input layer the size of the embedding and the hidden layers size 512, 256 and 128 respectively (with the exception of TM-Vec embeddings, where the model had only three layers and an input layer size matching the size of the 512-dimension embeddings). The layers were trained with 20% dropout and ReLU activation. The output layer was the same size as the number of functional categories being predicted and had a softmax activation.

These new models were used to make predictions on labeled sequences that were left out of the training sets. The number of correct predictions, or true positives (TP) was measured versus the false positives (FP), true negatives (TN) and false negatives (FN). Evaluations for the classifiers were measured per functional category using area under the receiver operating characteristic curve (AUROC), area under the precision-recall curve (AUPRC), and F1 scores (calculated using the following formula). For the PHROGs five-fold cross validation, true labels were taken from the database. ROC, PRC, AUC and F1 scores were calculated using scikit-learn `https://scikit-learn.org/stable/index.html` methods roc_curve, precision_recall_curve and auc. The F1, precision and recall scores were compared across models via boxplot.

### 6.3 TRAINED MODEL PERFORMANCES ON EFAM

A multi-class classifier was trained on the embedded EFAM database for each of the five model embeddings, using the same model architectures and training parameters as the PHROGs classifiers.

Each of the models was used to embed the EFAM database. As ground truth for making predictions, the HMM profiles in the EFAM database were annotated to the ten PHROGs functional categories using profile-profile HMM matching from hhsearch. The profiles were taken from Flamholz et al. An EFAM family was given a PHROGs family label assignment if the family matched a PHROGs HMM with an e-value ¡ 1E-10. The annotated HMM profiles were taken from `https://pubmed.ncbi.nlm.nih.gov/37205395/`.

The models were used to make predictions on the EFAM clusters. The number of "correct" predictions were counted and used to calculate the precision and recall scores for each model. Evaluations for the classifiers were measured per functional category using AUROC, AUPRC and F1 scores. The mean predicted value is the probability of a EFAM family matching its predicted functional category, with 0 indicating no probability and 1 indicating a 100% probability. The number of predictions were counted and plotted in a histogram by their mean predicted value.

Using the test set from each of the models, a per-class calibration analysis was performed using the scikit-learn calibration_curve method. EFAM VPFs with matches to annotated PHROGs HMMs were used to test the performance of the newly trained models. A perfectly trained model (where the mean predicted value is equivalent to the number of positives) is represented by the line in the middle of the plot. Models that are overconfident trend above the line, and models that are under confident trend below the line. Then, the calibrated classifier was used to predict EFAM VPFs not captured by PHROG HMMs.

To determine the biological relevance of these models, their F1 scores were plotted against their false discovery rates (FDR). The FDR threshold was determined to be 10%, following with the FDR threshold from Flamholz et al. (2024).

To determine whether the five models could make accurate functional predictions of biological interest, the integration and excision category was closely examined. This category was also selected so that the results from the model in Flamholz et al. (2024). could be compared with the results from the five models. The probability of a family that was predicted as "integration and excision" was plotted against the average protein length in that family.

## 7 REPRODUCIBILITY STATEMENT

The code used, instructions for running the code and required files for replicating the experiment are stored on an Anonymous Github `https://anonymous.4open.science/r/plm-model-comparison-7733/README.md`. Links to experimental data are also present in this repository.

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

# A    APPENDIX

| Supplementary Table 1. | | |
|---|---|---|
| Model | Average AUPRC | Average AUROC |
| ProstT5 | 0.6950905879 | 0.9245461656 |
| ProtT5 | 0.7181992387 | 0.9283766522 |
| TM-Vec | 0.6926709991 | 0.9165514403 |
| CARP | 0.6950327957 | 0.9193346538 |
| ESM-2 | 0.6633656441 | 0.9118546932 |
| Transformer_BFD | 0.6285752236 | 0.9032367544 |

**Supplementary Table 1.** PHROGs Classifiers Average AUROC and AUPRC. The average AUCs for the receiver operator characteristic and precision recall curves across all categories were calculated and outputted below.

| Supplementary Table 2. | |
|---|---|
| Model | Average AUPRC |
| ProstT5 | 0.9620782614 |
| ProtT5 | 0.9597163143 |
| TM-Vec | 0.9533969517 |
| ESM-2 | 0.9455279952 |

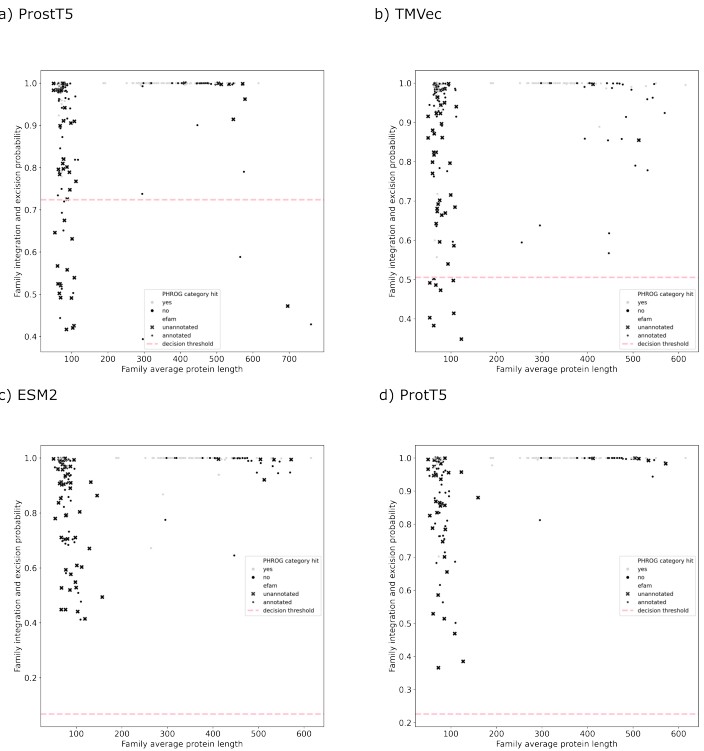

**Appendix 1.** EFAM Classifier Integration and Excision Prediction Probability as a Function of Average Protein Length. All of the EFAM VPFs that were predicted as integration and excision had probabilities that correlated with the average protein length in a family. EFAM VPFs that do not match PHROG HMMs and are unannotated in EFAM are labeled with (x). The decision threshold was determined from the maximum F1 threshold from the integration and excision category.

