# OpenReview forum: "Comparing Protein Language Models Using Remote Homology Detection for Phages"
_ICLR.cc/2025/Conference — ICLR 2025 Conference Withdrawn Submission_

### Official Review · Reviewer_8kFM · 2024-10-23

**Soundness:** 3
**Presentation:** 2
**Contribution:** 2
**Rating:** 3
**Confidence:** 5

**Summary:**

In this work, the authors compared several protein language models in a difficult remote homology detection task for phage proteins, to investigate the strengths and limitations of current PLMs in biological applications. They evaluated the results of different models and analyzed the factors that influence model performance. Finally they concluded that many PLMs are useful and incorporating specific domain knowledge could boost model performance.

**Strengths:**

The authors proposed an important question that how can biological researchers choose a PLM for their specific use cases,  since there are more PLMs publicly available. Also the remote homology detection is an important biological problem for researchers. It is a good attempt to evaluate PLMs' performance on viral sequence databases.

**Weaknesses:**

**1. Insufficient contribution**

This work seems to be an empirical study that evaluates several existing PLMs on  the remote homology detection task. However, many other powerful PLMs, some of which are especially for remote homology detection, are not included in this investigation, e.g. ProTrek[1], PLMSearch[2], DHR[3], etc. Besides, the authors used ESM-2 150M instead of ESM-2 650M due the memory limitation. The ESM-2 650M is commonly thought to be the best in ESM-2 series. Analysis based on only 5 models may not be that helpful and we need a more comprehensive evaluation.

The analysis is not sufficient. The authors advised to use models with large, diverse training datasets and structure-based objectives. However, they didn't further explore how these factors affect final performance. For instance, they could evaluate the sequence overlap between the test viral databases and different training datasets to investigate different models' knowledge regarding these viral sequences. The analysis and conclusion in the paper may not be helpful enough for biological researchers to choose a good PLM for their tasks.

**2. Paper writing**

The paper writing is confusing. I didn't see a figure to exhibit overall pipeline of model evaluation, which is hard for readers to quickly understand the outline about the work. The method part lies at the end of the paper, and the classifier architecture is described by texts. In part 3 the authors introduced all five models, which I think would be better  to put them in the appendix. In the Table 1 (the first table does not have caption), there seems to be a typo in the number of parameters for ProstT5 (17M looks like the dataset size).

*[1] ProTrek: Navigating the Protein Universe through Tri-Modal Contrastive Learning*

*[2] PLMSearch: Protein language model powers accurate and fast sequence search for remote homology*

*[3] Fast, sensitive detection of protein homologs using deep dense retrieval*

**Questions:**

Please see the weaknesses above.

---

### Official Review · Reviewer_4nx3 · 2024-11-04

**Soundness:** 2
**Presentation:** 2
**Contribution:** 2
**Rating:** 3
**Confidence:** 4

**Summary:**

The study compared five popular protein language models(pLMs) including ProtT5, ProstT5, TM-Vec, ESM-2, and CARP for performance on remote homology detection. The authors used two  different datasets(PHROG and EFAM) to evaluate the aforementioned pLMs. The authors found that pLMs are good at the given task when sufficient data is given and transformer based models are in general better than other architectures.

**Strengths:**

1. The authors covered a wide range of pLMs across different model architecture and training objectives to get a more comprehensive view of performance landscape.
2. The task used in this study is highly ralavent for many biologists that are interested in the use of pLMs in their research and provided a good guidance on model selection.
3. Highly reproducible result with shared codebase.

**Weaknesses:**

1. The title suggested remote homology detection which generally refers to structural homologues with low sequence identity. However, in the dataset preperation, the authors didn’t seem to account for the sequence similarity which is crucial in evaluating any biological sequence model performance.
2. The authors picked 5 models from 5 different pLMs which varies widely in terms of model sizes(17M - 3B parameters) with diverse model architectures. It doesn’t offer clear comparison and numbers shown are not very meaningful for a benchmark point of view. A better approach should be comparing various model architectures with similar model size or  various model size within the same model architecture.
3. The lack of naive benchmark and ab initio models trained from scratch or SOTA models for the given task. Those benchmarks will put the model tested in context which is extremely important  for a benchmarking effort.

**Questions:**

1. Why not sequence similarity filters are applied in the dataset construction? The term “remote homology” is used rather loose here.  What’s the definition of remote homology was used in this study? Is it consistent with what others have used it for?
2. What’s the rationale of you model selection? Those 5 models compared are qutie different in both model size and architecture, this makes the numbers in the paper not very comparable which might confuse your authors. The performance is worse for the 150M ESM2 models compared to a 3B ProtT5-XL model, how do they compare if a 3B ESM2 is used instead? Regardless, the choice of 150M ESM2 model is quite odd considering the most widely used ESM2 model is the 650M one to balance model performance and efficiency.

---

### Official Review · Reviewer_jn9t · 2024-11-04

**Soundness:** 1
**Presentation:** 1
**Contribution:** 1
**Rating:** 3
**Confidence:** 4

**Summary:**

This paper compares different protein language models (pLMs) in the challenging context of remote homology detection. The authors demonstrate that pLMs are valuable tools for remote homology detection in phage genomes, providing a useful framework for biologists to integrate these models into their experimental analyses.

**Strengths:**

This paper provides code and notebooks to facilitate other research scientists. It has good reproduction.

**Weaknesses:**

However, remote homology detection is a well-established task with numerous existing methods, including BLAST, HMM-based models, alignment-based deep learning, and similarity-based spatial distance approaches. The paper does not review or compare these established methods, which limits the depth of the analysis. Additionally, while the authors just pre-train and fine-tune the pLMs on relevant datasets, the overall concept presented lacks novelty and robustness.
Other issues with the article include:
1. The abstract should not contain citations, and its format does not adhere to standard conventions.
2. The subfigures in Figure 1 are too small to read effectively.
3. The legend under Figure 4 does not specify 'Figure 4', which could confuse the reader regarding its reference in the main text.
4. The label fonts in Figures 3b and 5 are too small, while those in Figure 3a remain legible.
5. Table 2 in the appendix lacks a title and description.

**Questions:**

As shown in the weaknesses.

---

### Official Review · Reviewer_rGYT · 2024-11-05

**Soundness:** 2
**Presentation:** 1
**Contribution:** 1
**Rating:** 3
**Confidence:** 5

**Summary:**

This paper compares five protein language models on remote homology detection tasks for bacteriophage proteins. Their finding is that transformer-based models with structure-based training on diverse datasets generally outperform others. Authors show that demonstrate models are useful for remote homology detection in phage genomes.

**Strengths:**

It is important to benchmark PLMs in different tasks. In that sense, the paper address a noteworthy task.
Results demonstrate that the trained database has an impact. Models trained on more diverse datasets perform better, which is an interesting observation.

**Weaknesses:**

- The technical contribution is limited.

- The authors limited their comparison to 5 PLMs. Many recent noteworthy PLMs could be included.  ProGen, Ankh, Tape. Also, it would be interesting to see if ESM3 performs better than ESM2.  These expanded comparisons could enhance the study's impact.

 - The approach of using precomputed embeddings with fine-tuned classifier layers is valid, yet exploring alternative fine-tuning strategies could provide more comprehensive insights. For instance, fine-tuning the entire model or the last few layers on the same dataset, particularly for models like ESM2, might reveal whether more extensive tuning impacts performance. Did computational constraints limit the ability to explore full model fine-tuning?

- Authors report ESM2’s relatively poor performance. The authors used the 150M ESM2 model rather than the larger version. It would be beneficial to discuss any potential limitations of using the smaller model and consider if these limitations might have impacted the conclusions regarding ESM2’s capabilities.

- The classifier is referenced from a previous paper without much detail in the current work. For clarity and completeness, a brief description of the classifier could be included.

- Figures 1 and 4 are difficult to read due to small font and marker sizes. Enlarging these elements and adjusting spacing could greatly improve figure legibility, ensuring that readers can fully engage with the visual data.

**Questions:**

How would a simpler baseline protein sequence encoding or a ProtVec representation perform on this task?

---

### Note · Authors · 2025-01-06

I have read and agree with the venue's withdrawal policy on behalf of myself and my co-authors.